# Medicinal Mushrooms as Multicomponent Mixtures—Demonstrated with the Example of *Lentinula edodes*

**DOI:** 10.3390/jof10020153

**Published:** 2024-02-15

**Authors:** Ulrike Lindequist

**Affiliations:** Institute of Pharmacy, Pharmaceutical Biology, University of Greifswald, D-17487 Greifswald, Germany; lindequi@uni-greifswald.de

**Keywords:** *Lentinula edodes*, shiitake, medicinal mushrooms, multicomponent mixtures, lentinan, eritadenine

## Abstract

Medicinal mushrooms are multicomponent mixtures (MOCSs). They consist of a large number of individual compounds, each with different chemical structures, functions, and possible pharmacological activities. In contrast to the activity of an isolated pure substance, the effects of the individual substances in a mushroom or its extracts can influence each other; they can strengthen, weaken, or complement each other. This results in both advantages and disadvantages for the use of either a pure substance or a multicomponent mixture. The review describes the differences and challenges in the preparation, characterization, and application of complex mixtures compared to pure substances, both obtained from the same species. As an example, we use the medicinal and culinary mushroom *Lentinula edodes*, shiitake, and some of its isolated compounds, mainly lentinan and eritadenine.

## 1. Introduction

In the history of mankind, plants and fungi were for a long time the only available means for the treatment of health problems. They were used in the form of whole plants and mushrooms or extracts prepared therefrom. Beginning in the 18th and 19th century, as individual constituents began to be increasingly isolated from plants and fungi, there has been increasing interest in the chemical composition of herbs and their constituents. A breakthrough was the extraction of morphine in 1806 by the pharmacist F.W. Sertürner (1783–1804) from opium (*Papaver somniferum*) [1]. This was quickly followed by other discoveries of active plant and fungi substances [2,3,4,5].

As a result, complex herbal and fungal preparations were increasingly replaced by isolated or synthesized pure substances. They were easier to dose and had often a stronger and better predictable effect than complex mixtures.

Nevertheless, plants, fungi, and, to a lesser degree, animals have retained their importance as biogenic drugs. In some regions of the world, they are still the only or nearly only source for medicinal treatment [6]. The Traditional Chinese Medicine (TCM) and other medicinal systems mainly from East Asia strongly rely on mixtures of several plants and/or fungi [7,8].

Apart from the isolation of single substances, complexly composed products, so-called multicomponent mixtures (more than one constituent substances, MOCSs), are obtained from the natural materials. A multicomponent mixture contains numerous individual constituents with different functionalities. Together, they contribute to the properties of the complete mixture. Typical MOCSs are extracts of herbal (including fungal) drugs, essential oils, or animal venoms [9,10].

Today, it has become more and more evident that MOCSs prepared from medicinal plants could have advantages in comparison to pure substances. The use of an essential oil that contains several hundred individual constituents, e.g., in aroma therapy or to treat respiratory diseases, local infections, or digestive disorders, is much more commonplace and beneficial than the use of a single substance isolated therefrom [9,10].

By using the example of *Lentinula edodes* (Berk.) Pegler, shiitake (Figure 1), it is shown that this is the case not only for plants but also for medicinal mushrooms. Shiitake mushrooms have been used medicinally in East Asia for more than 2000 years. Because of their very good culinary properties, they are popular mushrooms worldwide. Shiitake mushrooms are the second-most cultivated mushrooms worldwide, accounting for about 25% of global edible fungus production [11]. *L. edodes* and some of its constituents are well known for their health-promoting properties and therapeutic potential. Lentinan, a polysaccharide isolated from the mycelium, is used as a single compound as part of an integrated tumor therapy [12,13,14,15].

## 2. MOCSs versus Single Compound in General

The diversity of molecules with different chemical and physical properties and various pharmacological activities and their interplay is the base for the multi-target activity of an MOCS compared to the one-target activity of a single compound. This means that MOCSs often exhibit a broad spectrum of biological activities and can be more useful for homeostasis of the human body than a single compound [9,10,16,17].

The individual compounds present in the mixture may enhance each other (synergism), complement each other (additive effect), or attenuate each other (antagonism). The resulting effect may be not only positive, but may also compensate for, attenuate, or cancel out possible undesirable effects of other components [9,10,16]. MOCS components may modulate their mutual resorption but may also interact with specific compounds ingested from food or medicines and may also affect typical characteristics of smell and taste [18,19].

A pure substance present in the same amount as in natural source mixtures usually does not reach the same pharmacological effects on a quantitative and qualitative scale. However, as part of a natural mixture, the same chemical constituent may exhibit a clear biological activity [9].

The combined effect of many compounds acting together is called the entourage effect. It means that several different compounds act together to achieve an effect that would be impossible for them to achieve on their own. The entourage effect is particularly well known for the diverse ingredients of *Cannabis sativa*, hemp, and their pharmacological activities [20,21,22].

In addition to the multi-target effect, MOCSs have further advantages in comparison to single substances. In most cases, the costs and the ecological impact of their production are lower.

Of course, there are also disadvantages. The more complex an MOSC is, the more difficult it is to verify its authenticity and genuineness. Higher challenges in quality control and standardization result. The possibilities for application are limited. While peroral and external application do not pose a problem, parenteral administration is largely ruled out. The multitude of effects is difficult to detect in clinical trials and the mechanisms of action often remain unclear.

The conclusion can only be that both single substances and MOCSs are necessary and justified. For example, while a pure antibiotic is essential for acute pneumonia, a herbal or fungal preparation may be sufficient and even better for a mild cold. In many cases, a combination makes sense and is in line with the wishes of many patients [23].

## 3. Chemistry and Biological Effects of Single Compounds from *Lentinula edodes*

### 3.1. Overview

Table 1 gives an overview of the nutritionally relevant ingredients of *Lentinula edodes*, and Table 2 summarizes those substances that are primarily of pharmacological interest. However, the transitions between these two groups are fluid. For example, dietary fiber is a bulk-giving component of the diet that contributes significantly to eliminating the feeling of hunger. On the other hand, dietary fiber helps to promote peristalsis and maintain a healthy microbiome and can therefore also be regarded as a pharmacologically active ingredient. Another example is ergosterol, which itself is hardly pharmacologically active, but can be converted to the pharmacologically active ergocalciferol (vitamin D2).

Among the nutritionally important characteristics worth highlighting are the high content of dietary fiber and of proteins containing all essential amino acids, the presence of vitamins and minerals, and the low energy density. Also worth noting are the pleasant sensory properties for which the volatile compounds are mainly responsible [14,15,24,32,34,35,36].

### 3.2. Lentinan

The best studied biologically active substance is the polysaccharide lentinan (Figure 2). This glucan consists of β-1,3-linked glucose units to which β-1,6 side chains are attached after every five linear glucose residues. The secondary structure is probably a curdlan-type helix conformation, which can be easily converted into a triple helix by changing the external conditions. The average molecular mass is 5.2 to 6.9 × 10^4^ Daltons. Lentinan can be isolated from hot water extracts of *L. edodes* by a sequence of fractional precipitation and chromatographic purification steps [12,25,37,38].

Glucan molecules such as lentinan are considered to be pathogen-associated molecular patterns (PAMPs) targeting pattern recognition receptors (PRRs). They target dectin-1, complement receptor 3 (CR3), and further receptors on the surface of monocytes, macrophages, and other immune cells and are capable of stimulating both innate and adaptive immune responses (biological response modifier, BRM). Thus, numerous signaling pathways are activated. For example, this can lead to the release of cytokines, the strengthening of phagocytosis, and an oxidative burst response [39]. As a result, the host’s own immune defense against tumor cells can be stimulated. When applied parenterally (0.5–1 mg/day) or orally, lentinan can be used as part of an integrated tumor therapy together with chemotherapeutics and other measures (operation, radiation).

The immunostimulating effects of lentinan have been justified in numerous in vitro and in vivo investigations. For example, such studies show a stimulation of the release of various cytokines (IL-1α, IL-2, IFNγ, TNF α) and of the activity of natural killer cells (NK cells). In animal experiments, an inhibition of tumor growth could be achieved. Clinical studies and observations show an improvement in life quality and a prolongation of survival time of patients with mainly gastrointestinal tumors that had taken lentinan additionally to chemotherapy [14,39,40,41,42,43].

In the context of the CoV-19 pandemic, the antiviral activities of lentinan and other mushroom polysaccharides have gained increasing interest. The effects are mainly due to the immunostimulating properties of the compound [44]. β-Glucans derived from mycelia reduced cytokine levels involved in cytokine storms during CoV-19 infections and reduced lung infection [45]. Clinical studies of lentinan in HIV-positive patients showed promising results [46]. Antiviral effects against bovine herpes virus type 1, HIV, and HTLV-1 have also been shown for other polysaccharide preparations from *L. edodes* [14].

With regard to the other polysaccharides, it has to be mentioned that the extraction method and the extracted mushroom part (fruit bodies, mycelium, or spores) affect monosaccharide composition, molecular weight, degree of branching, and conformation and thus the biological activity of the compounds [39].

### 3.3. Eritadenine

Another well-studied constituent of *L. edodes* is the purine derivative eritadenine (Figure 3). The estimated content is between about 300 and 650 mg/100g DW. It decreases during cooking [29,30,31].

In animal experiments, it lowers cholesterol and triglyceride levels by another mechanism than the well-known statins. Instead of the inhibition of HMG-CoA reductase as statins do, it modulates the expression of genes for enzymes of lipid metabolism and promotes the uptake of LDL into liver cells [47,48,49]. Additionally, eritadenine inhibits the activity of angiotensin-converting enzyme (ACE) in vitro and thus has possibly antihypertensive effects in vivo [30]. By the inhibition of *S*-adenosyl-L-homocysteine hydrolase, it reduces the high blood level of the amino acid homocysteine, a risk factor for cardiovascular and neurodegenerative diseases [49,50]. All these activities together can be expected to have positive effects in the prevention of atherosclerosis and other cardiovascular diseases.

Positive effects on lipid levels could also be achieved with isolated polysaccharides. In rats that were fed a lipid-rich diet, the polysaccharides reduced total cholesterol, triglycerides, and LDL cholesterol. The expression of the adhesion molecule VCAM-1mRNA in aortic endothelial cells that promotes the development of atherosclerotic plaques was decreased and the activity of antioxidative enzymes increased [51].

A third remarkable substance that should be mentioned is the amino acid ergothioneine (2-mercaptohistidine-trimethylbetain). Its content is strongly dependent on the culture conditions and estimated to be between 1.22 and 198 mg/100g DW. Ergothioneine has been proposed to be an adaptive antioxidant that may protect against the tissue damage implicated in several chronic diseases and promote healthy aging [28].

Among the other constituents, phenolics and volatiles (15 alcohols, 13 aldehydes, 9 alkanes, 5 sulfur-containing compounds, and other, [31]) should be mentioned.

Untargeted metabolomics of 7 mushroom species including *L. edodes* showed that 1344 compounds were detected in all 7 investigated species. A total of 472 compounds were unique to shiitake; 126 of them were annotated and 346 were not annotated [28]. This means that there is yet a vast unexplored field.

## 4. Biological Effects of MOCS from *Lentinula edodes* (Extracts, Dried, or Fresh Mushrooms)

### 4.1. Immunomodulating Effects

In a randomized clinical study, healthy young adults (*n* = 52) consumed either one (5 g) or two (10 g) servings of dried fruit bodies of *L. edodes* daily for 4 weeks. Several immune parameters, including the levels of sIgA, interleukin-4, and interleukin-10 and the ex vivo proliferation of NK-T-cells, were upregulated. It was concluded that regular consumption resulted in strengthened immunity [52].

A standardized extract of mycelia elevated the level of IgG and some cytokines and the phagocytic index in children with refractory epilepsy. Thus, the immune status and the quality of life of the children were improved [53].

An ethanolic extract of fruit bodies decreased symptoms of experimentally induced atopic dermatitis. The effects were attributed to the contained polyphenols [33].

The immunomodulating effects can be exploited also in veterinary medicine. As an example, hot water and methanolic extracts of *L. edodes* fruit bodies improved the immune defense of poultry against infections with *Eimeria* Schneider, pathogens of coccidiosis, and other parasites [54].

### 4.2. Antitumor Effects

In contrast to the numerous studies with isolated lentinan for the treatment of tumor patients (see Section 3.2), there are only a few studies in which complex preparations were used. They were carried out with LEM, a mycelial extract consisting of lignin (80%), carbohydrates (10%), and protein (10%), besides a complex known as EP3 immunoreactive.

A small clinical study (*n* = 7) gave advice that the orally applicated LEM, given additionally to chemotherapy, increases several immune parameters and could possibly improve the quality of life of patients with mammary or gastrointestinal tumors [55].

An open pilot study suggested that a combined treatment with LEM and immunotherapy might improve quality of life and immunological function in cancer patients [56].

It can be assumed that the observed effects are due to high-molecular components including lentinan. The cytotoxic protein ledodin is likely not absorbed in an intact form when administered orally and probably does not play a role.

### 4.3. Effects-Reducing Risks for Cardiovascular Diseases (Lipids, Homocysteine, Blood Pressure, Redox State)

The first indications of antilipidemic effects date back to the 1970s. A daily intake of 90 g of fresh shiitake, 9 g of dried shiitake, and 9 g of UV-irradiated dried shiitake for 7 d lowered the mean cholesterol levels in young women by 12%, 7%, and 6%, respectively. In people over 60 years of age, the effects of all three diets were the same at 9% [57].

The effects were analyzed in more detail in animal studies and in further clinical studies, as described shortly.

An amount of 1, 2, or 4% of LEM in rabbit food enriched with cholesterol (1%) for 8 weeks reduced in a non-dose-dependent manner the number of foam cells in the aorta of rabbits in comparison to the control [58].

Amounts of 5, 10, and 20% of *L. edodes* powder in a high-fat diet of mice for 4 weeks led to a dose-dependent reduction in the serum levels of total cholesterol, LDL, and triglycerides in comparison to mice that had been fed the high-fat diet alone. Fat accumulation in the liver and the formation of atherosclerotic plaques were reduced. The mRNA expression of cholesterol-7-α-hydroxylase was decreased in hypercholesterolemic mice and increased in the shiitake groups. Eritadenine alone (10 mg/kg body weight in high-fat diet) showed comparable effects [59].

*L. edodes* powder (5, 10, and 20%) in the diet of homocysteinemic mice for 2 weeks reduced the increased hepatic *S*-adenosyl-L-homocysteine hydrolase level and restored the mRNA expression levels of DNA methylases that were reduced owing to homocysteinemia. The same effect was achieved by feeding on pure eritadenine [60].

A water extract from *L. edodes* containing water-soluble α- and β-glucans and fucomannogalactans reduced triglyceride levels in the liver of normo- and hypercholesterolemic mice and modulated the transcriptional profile of some genes involved in the cholesterol metabolism [61].

A β-D-glucan-enriched mixture obtained from fruit bodies (10.4 g/day, corresponding to 3.5 g β-D-glucans/day), incorporated in food products, was given to hypercholesterolemic patients. The randomized, placebo-controlled clinical study with 52 participants (verum + control) found changes in the colonic microbiota in the mushroom group. But there were no differences regarding lipid-related parameters [62].

In a randomized, double-blind, placebo-controlled study, individuals with borderline hypercholesterolemia received shiitake bars. Participants in the intervention group showed a 10% reduction in triglycerides after 66 days of consuming the bars. Their redox status was improved [63].

Antioxidative effects have been shown for aqueous and methanolic extracts of fruit bodies, fractions of polysaccharides, and mycelial preparations [11,14,64]. They correlate with the phenolic content of the mushrooms [10] and can also be due to ergothioneine.

The various effects that reduce the risk of cardiovascular diseases can be attributed to eritadenine, phenolics, ergothioneine, and possibly also to high-molecular components.

### 4.4. Osteoprotective Effects

Osteoprotective effects have been detected in vitro and in animal studies. They affect both the bone-building osteoblasts and the bone-degrading osteoclasts.

A water extract of the fruit bodies of *Lentinula edodes* (<30 µg/mL) stimulated the activity of alkaline phosphatase (ALP) and the mineralization in two human osteoblastic cell lines (HOS 58 and SaOS-2) [65]. Similar effects and an increase in osteocalcin production have been found in the mouse osteoblast cell line MC3T3-E1 and in primary rat osteoblasts. The bone-decreasing activity of rat osteoclasts was inhibited [66].

In another investigation, a water extract and its subsequent ethyl acetate fraction suppressed the differentiation of bone-marrow-derived macrophages from mice to osteoclasts. The fraction (10 µg/mL) inhibited the RANKL-induced differentiation of osteoclasts by blocking NFATc1. Comparative transcriptome analysis revealed that the fraction downregulated a set of RANKL-associated target genes, including *Nfatc1,* and inhibited the expression of this gene. In a zebrafish model of glucocorticoid-induced osteoporosis, the fraction rescued an osteoporotic phenotype [67].

A β-glucan from *L. edodes* (100 to 400 µg/mL) increased the ALP activity of MC3T3-E1 cells but, alone, had no influence on the mineralization of the cells. Such an effect was only observed in combination with BMP-7 (bone morphogenetic protein) [68].

A combination of water extracts of *L. edodes* and *Grifola frondosa* (Dicks.:Fr.) Gray, maitake, is already on the market as an osteoprotective food supplement. The peroral administration of the product to ovariectomized rats for 42 days significantly reduced trabecular bone loss at the lumbar spine in the animals [66].

Until now, it remains unclear which constituents are responsible for the effects of *L. edodes* on bone cells.

### 4.5. Antimicrobial Effects

The antimicrobial effects of *Lentinula edodes* are directed against bacteria, fungi, and viruses or they are based on the reduction in biofilm formation.

Aqueous and ethanolic extracts of fruit bodies as well as the liquid cultivation medium exhibited in vitro antibacterial activity against a broad spectrum of mainly Gram-positive bacteria [14].

The antiviral effects of crude aqueous or ethanolic extracts were found in vitro, e.g., against poliovirus type 1 (PV-1), bovine herpes virus type 1 (BoHV-1) [69], herpes simplex virus-2 (HSV-2) [63], and SARS CoV-2 [70]. The mycelial preparation LEM inhibits influenza virus by interfering with virus entry into the host cells and activation of the immune response through the type I IFN pathway [71]. Antiviral activities against HIV, e.g., by inhibiting HIV-1 reverse transcriptase, and against hepatitis B and hepatitis C virus were found in in vitro studies for LEM, further polysaccharides, the protein lentin, lignin, and laccase (for overview [72]).

In vivo, the antiviral activities are, at least partly, due to the immunomodulating effects of *L. edodes*.

This is supported, e.g., by the investigations with AHCC^®^. AHCC (active hexose-correlated compound) is a standardized extract of mycelia, composed of 74% of the dry weight of polysaccharides, of which 20% is a partially acetylated α-1,4-glucan. The intake of AHCC improves human health and increases animal immunity against various viral diseases by modulating NK cells and subsets of T cells [73]. AHCC modulates numbers and functions of immune cells including NK cells and T cells; supplementation in defending the host against infections and malignancies is recommended [74].

Shiitake extracts reduce oral biofilm formation and have anti-gingivitis and anti-caries effects [64]. In a short-term clinical study, a low-molecular-weight fraction of the mushrooms showed anticaries effects. However, they were lower than those of the positive control meridol (amine fluoride/tin fluoride solution) [75,76].

### 4.6. Cosmetics

An ethanolic extract containing polyphenols and β-glucan inhibits the expression of MMP-1 and MMP-9 and increases the expression of type I procollagen in UVA- and UVB-irradiated HaCaT keratinocytes. The results suggest that *L. edodes* may be developed as a cosmetic material to suppress UV-exposure-mediated skin aging [77].

In another study, an ethanolic extract of *L. edodes* and a cosmetic cream formulation incorporated with the extract inhibited the production of the inflammatory mediator nitric oxide (NO) by lipopolysaccharide (LPS)-stimulated macrophages and thus showed anti-inflammatory activity. In addition, both the extracts as the cream exhibited antibacterial effects against *Enterococcus faecalis* and methicillin-sensitive and methicillin-resistant *Staphylococcus aureus* [78].

### 4.7. Toxicological Aspects

Ledodin is a 22 kDa protein consisting of 197 amino acid residues. It strongly inhibits protein synthesis in a cell-free rabbit reticulocyte lysate system. It possesses *N*-glycosylase activity on the sarcin-ricin loop of the 28S rRNA of mammalian ribosomes. Unlike ribosome-inactivating proteins (RIPs) from plants, it is inactive against ribosomes from insects, fungi, and bacteria [26]. It can be assumed that the large molecule is not adsorbed in an intact form after peroral application, so it should be not of concern for the safety of the consumed mushrooms or mushroom preparations.

Especially when raw or uncooked mushrooms are consumed, linear flagellated dermatitis (shiitake dermatitis), pruritus, fever, diarrhea, and mucosal ulcers can develop. They are attributed to lentinan [79]. In this case, it is necessary to avoid the contact with and the consumption of shiitake.

Due to its rich content of insoluble fiber, digestion may be difficult. This may lead to luminal obstruction when very large pieces of the mushrooms are ingested. Cases of intestinal obstruction secondary to shiitake mushroom intake that required surgical intervention have been described [80].

## 5. *Lentinula edodes*: MOCS versus Single Compound

A comparison of the effects of single substances isolated from *Lentinula edodes* with those of MOCSs of the mushrooms (extracts, mushroom powder, fresh or prepared mushrooms) clearly shows that the MOCSs have a much broader spectrum of activities (one target–multi target). For example, an aqueous extract exerts antilipidemic, osteoprotective, and antimicrobial effects (see Section 4.3, Section 4.4 and Section 4.5). It is still unclear which effects can be assigned to which ingredient, and it can be assumed that only the interaction of several components results in the effect pattern. The combination of effects offers excellent opportunities for the prevention and control of civilization diseases such as cardiovascular diseases and age-related problems.

In the case of MOCSs, both mushroom extracts and powders as well as the mushrooms themselves used for culinary purposes have their place in the prevention and therapy of health problems. Extracts, whose composition can be controlled and in which the desired bioactive compounds are enriched, are mainly used when specific therapeutic or preventive purposes should be achieved. In mushroom powder, the natural mixture of components is not changed. They are considered for preventive purposes. Prepared mushrooms are used primarily for culinary enjoyment. The high content of insoluble fiber in mushroom powder and mushroom food and the content in ergosterol as a precursor for vitamin D2 should be emphasized. Mushrooms are the only non-animal-based food containing vitamin D precursors, and hence they are the only natural vitamin D source for vegetarians [81]. Nondigestible components like chitin can support a healthy gut microbiota [82] and promote digestion.

In contrast to ingesting single substances, there are many more possibilities to increase the variability of MOCSs. This already begins with the cultivation. The variation in cultivation conditions allows the diversity of products to be changed and increased. The concentration of primary metabolites was high in *L. edodes* grown on sawdust, which produces a high growth rate. In contrast, log-cultivated *L. edodes*, which were similar to wild mushrooms, had high levels of bioactive compounds and high antioxidant activity [11]. The addition of suitable substances, e.g., zinc or selenium or phenolic acids, to the cultivation substrate leads to enrichment of the substances in the cultivated mycelium and thus to better biological properties, e.g., stronger anti-inflammatory effects [83]. Exposition of the mushrooms, e.g., during cultivation or later, to UV-B light increases the vitamin D content [84].

Another important decision is the choice of mushroom material to be processed: fruit body or mycelium? This question cannot be answered in general but must be answered on a case-by-case basis.

By selecting the extraction conditions (solvent, temperature, time, further treatment processes), it is possible to enrich individual components. For example, the glucan content can be increased by hot aqueous extraction. Eritadenine and phenolic ingredients are enriched in aqueous ethanolic extraction. As the insoluble fibers remain in the residues, the bioavailability of the active ingredients is improved. The extract yield can be increased by additional technical measures, e.g., pulsed electric field treatment [85].

In the case of mushrooms as food, cooking methods influence the content. For example, steaming, boiling, and air frying for 5–20 min decrease the content of crude polysaccharides and eritadenine in the mushrooms. The lowest losses of crude polysaccharides and eritadenine were observed for oven baking for 5 min and 15 min, respectively [31].

The analytical requirements to ensure the quality of an MOCS are much higher than for a pure substance and, in the case of fungi, still a major challenge. A comparison of different methods for glucan determination in *Lentinula edodes* (Megazyme kit, Glucatell^®^, Aniline blue-fluorometric method) yielded completely different results [86].

Pharmacologically, the investigation of an MOCS with a broad spectrum of activities and changing composition is more challenging than with a single compound. At present, we are far from knowing all the effects variables can have. Until now, many activities have been demonstrated only in vitro. What impact do the mushrooms have on the microbiome and how does that affect health? Are there interactions between the constituents of the MOCSs and between the MOCSs and approved drugs? Which components are really responsible for shiitake dermatitis and is it possible to remove them?

From an economic and environmental point of view, the use of fresh or dried mushrooms is more favorable than the extraction and the isolation of pure substances, which require additional energy input and often organic solvents.

From a regulatory standpoint, the isolated substances have the more likely chance of being approved as drugs. For lentinan, this is the case in some Asian countries. Mushrooms cultivated for food purposes are, of course, subject to food legislation. In Europe, most products made from medicinal mushrooms are marketed as food supplements. They are therefore also subject to food legislation and may not have any indication-related advertising [87]. The European Food Safety Authority (EFSA) has licensed a sterile aqueous mycelial extract of *Lentinula edodes* (Lentinex^®^) as a novel food containing 0.8–1.2 mg/mL of lentinan, free glucose, proteins, and amino acids. The proposed daily intake is 2.5 mL containing 1 mg of lentinan (corresponding to 41.7 µg/kg body weight/day for a person with 60 kg body weight [88]). The product given orally to healthy adults was safe and induced an increase in the number of circulating B cells [89]. The detected improvement in well-being was higher for subjects who had lower initial well-being than for subjects with higher initial well-being [90].

## 6. Conclusions

Both isolated individual substances and MOCSs (extracts, powders, mushrooms prepared as food) have their justification. Single substances such as lentinan are used as approved drugs in a precisely dosed form and often parenterally for the therapy of severe diseases, in this case, tumor diseases. MOCS, mainly extracts of fruiting bodies, sometimes of mycelia, are used as (supportive) therapeutics and as preventive remedies. They reduce the risk of metabolic syndrome, cardiovascular diseases, infections, and osteoporosis and are therefore of great interest for the prevention of age-related health problems and so-called lifestyle diseases. In tumor patients, they can be supportive but not replace standard therapies, improve the prognosis, and increase the quality of life. Last but not least, shiitake mushrooms are not only an excellent tasting food to enjoy, but also provide humans with substances valuable to health, e.g., dietary fiber, essential amino acids, and precursors of vitamin D, even when used in this way.

## Figures and Tables

**Figure 1 jof-10-00153-f001:**
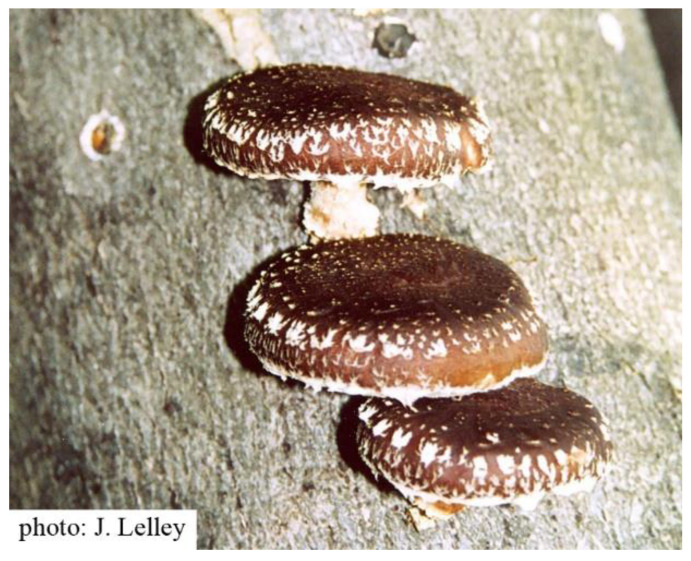
*Lentinula edodes* (photo: Prof. Jan Lelley, with permission).

**Figure 2 jof-10-00153-f002:**
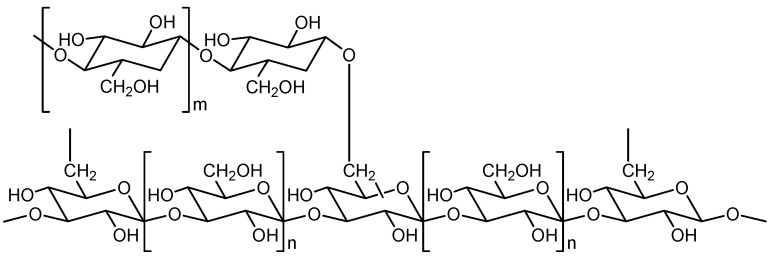
Partial structure of lentinan. n: the glucose unit in brackets is present n times in the 1,3-chain; m: the glucose unit in brackets is present m times in the 1,6-side chain.

**Figure 3 jof-10-00153-f003:**
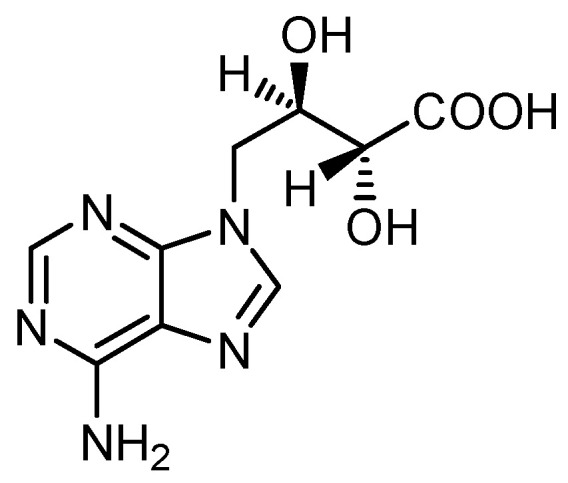
Structure of eritadenine.

**Table 1 jof-10-00153-t001:** Composition of *Lentinula edodes* fruit bodies under special consideration of nutrients (according to [24] and references therein).

Substance	Content	Remarks
Dry matter	8.4–20.2 g/100 g FW	
Total dietary fiber	32.4% of dry matter	high content of insoluble fiber including chitin
Ash	4.3–6.7 g/100 g DW	
Energy	33.2–77.2 kcal/100 g FW	
Crude protein	4.4–21.4 g/100 g DW	
Essential amino acids	32.0% of total amino acids	
Free amino acids	1.25–3.16 g/100 g DW	affect taste of fresh mushrooms
Crude fat	0.9–6.3 g/100 g DW	
Polyunsaturated fatty acids	82.0% of total fatty acids	
Carbohydrates	67.9–87.2 g/100 g DW	
Soluble sugars and polyols	Caps: 111.6 g/100 g DW Stipes: 125.0 g/100 g DW Dried powder: 38.3 g/100 g DW	
Ergosterol	217–495 mg/100 g DW	precursor of vitamin D2
Tocopherols	5.4–32.3 µg/100 g DW	
Vitamin C	25 mg/100 g DW	
Vitamin B1	0.6 mg/100 g DW	
Vitamin B2	1.8 mg/100 g DW	
Vitamin B12	0.8–5.6µg/100 g DW	
Folates	0.3–0.66 mg/100 g DW	
Niacin/niacinamid	31 mg/100 g DW	
5’-Nucleotides	2.42 g/100 g DW	
Monophosphates of 5’-guanosine, 5’inosine and 5’xanthosine	48% of 5’-Nucleotides	affect taste of fresh mushrooms

DW: dry weight, FW: fresh weight.

**Table 2 jof-10-00153-t002:** Components of *Lentinula edodes* with proved pharmacological activities and/or other special properties.

Substance Group	Substance	Activity and/or other Properties	References
Polysaccharides	Lentinan	Immunomodulating	[12,25]
Proteins	Ledodin	Atypical ribosome-inactivating protein; Cytotoxic	[26]
	Lentin, a 27.5 KDa protein	Antifungal; Inhibition of HIV-1 reverse transcriptase; Inhibition of proliferation of leukemia cells	[27]
Amino acids (additionally to proteinogenic amino acids)	Ergothioneine	Antioxidative	[28]
	γ-Amino butyric acid (GABA)	Neurotransmitter	[24]
Nucleotide derivatives	Eritadenine (lentinacin, lentysin); Desoxyeritadenine.	Antilipidemic; Inhibits angiotensin converting enzyme (ACE); Reduces homocysteine level	[24,29,30,31]
Cyclic-sulfur-containing compounds	1,2,4-Trithiolan, arising from the non-volatile lentinic acid; 1,2,3,4,5,6-Hexathiocycloheptan (hexathiepan); 1,2,3,5,6-Pentathiepan (lentionin); 1,2,4,6-Tetrathiepan.	Main odor substance	[12,24,32]
Phenolics	p-Hydroxybenzoic acid; Vanillic acid; Syringic acid; p-Coumaric acid; Further Polyphenols.	Antioxidative	[11,24,31,33]

In Table 2, content data are omitted since comparability is not given, due to different test materials and determination methods as well as often missing data in the literature.

## Data Availability

Not applicable.

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
