# Peer review of "Medicinal Mushrooms as Multicomponent Mixtures—Demonstrated with the Example of Lentinula edodes"

_jof, 2024, doi:10.3390/jof10020153_

Round 1
Reviewer 1 Report
Comments and Suggestions for Authors
This review manuscript is meaningful, comparing single substances with multicomponent mixtures (MOCS) derived from mushrooms. Despite some shortcomings of MOCS, it also has many advantages. The choice of topic for this manuscript is very good. It is recommended to accept after minor revisions. My comments are as follows:
1. The abstract needs to be improved.
2. Figures 2 and 3 need to be refined and indicate what each symbol means.
3. On line 279, please clarify the mechanism of laccase's antiviral properties.
Reviewer 2 Report
Overall good, only some formats need to be properly refined.
1. Line 94-97: It is recommended that Tables 1 and 2 be modified to three-wire tables.
2. Line113-114 &Line148-149: It is suggested to improve the polysaccharide structure diagram of Figure 2, because the current labeling cannot be recognized normally.The same problem exists in Figure 3.
3. Line391: delete (!).
Reviewer 3 Report
Comments and Suggestions for Authors
In the Review paper entitled: "Medicinal Mushrooms as Multicomponent Mixtures (MOCS) – Demonstrated at the Example of Lentinula edodes" the author gave the overview of active components from mushrooms, their advantages and disadvantages.
The paper is well thought out, still it requires improvements.
Line 8: The sentence: "They are multicomponent mixtures (MOCs)." have to be extended or attached to previous or the next sentence. Please revise this sentence.
Line26: Please add reference.
Line 33: Please add references.
Line 75: Reference is missing.
Line 95: DW and FW should be written under the Table.
Tables 2 and 3 are not properly formatted. All the lines are not necessary to be present in table. Please make it more appealing
Line 100: Correct the word worthighlighting.
Figure 2. Please change this Figure, it is poorly designed. The structure can't be seen from the Figure.
Lines 170-175: The sentences in this part should be better written.
Section 4.1.:This section have to be improved with more scientific information.
Line 180:Please, state which parameters.
Line 183:Please add which parameters.
Line 194: Please add a reference
Line 242: and instead of und
Section 4.6. In this section only one reference is given. There are much more research on this topic, so please improve this section.
Line 344: Delete the bracket
Line 347: The question: "Fruiting body or mycelium" is not answered in the following text. So the answer should be added or the question has to be deleted.
354: The content of what exactly?
Line 357: Please specify the methods and add the results from the reference 74
Comments on the Quality of English Language
Please uniform the passive voice in the text. It should be present simple or past simple passive voice. Not both.
There are many parts of the paper that require the improvement of sentences' construction.
Reviewer 4 Report
The MOCS (more than one constituent substances) concept has been recently dealt with in literature concerning herbs, whereas it is apparently missing in literature concerning mushrooms. From this point of view, the paper is original. On the other hand, the MOCS concept itself is not new at all and sounds like a “cool” phrase for a very old, well-known idea: by the way, this is not author’s fault.
I know the author has devoted many years in studying bioactive molecules from herbs and mushrooms and therefore feels confident with causes, effects, processes and molecules. However, she should keep in mind this is a review and every passage should be consequently provided with adequate references. Moreover, I suggest to cite other reviews as little as possible in order to avoid a “review of reviews” approach.
Based on the author’s experience, I also suggest to expand points concerning single molecules or categories of molecules and biochemical and/or physiological processes.
please see file in attachment

Round 2
Reviewer 3 Report
The Author has answered all questions and suggestions from review stage 1 and paper is now ready for publication.
Paper have been improved and now is fine for publication.